# Retronasal Aroma of Beef Pate Analyzed by a Chewing Simulator

**DOI:** 10.3390/molecules27103259

**Published:** 2022-05-19

**Authors:** Kazuhiro Hayashi, Yuji Nakada, Etienne Sémon, Christian Salles

**Affiliations:** 1Institute of Food Sciences & Technologies, Ajinomoto Co., Inc., Kawasaki 210-8681, Japan; yuji.nakada.d3a@asv.ajinomoto.com; 2Centre des Sciences du Goût et de l’Alimentation, CNRS, INRAE, Institut Agro, Université de Bourgogne-Franche Comté, 21000 Dijon, France; etienne.semon@orange.fr (E.S.); christian.salles@inrae.fr (C.S.)

**Keywords:** aroma release, pate, chewing simulator, fat content, elderly

## Abstract

In retronasal aroma, the targeted aroma compounds are released from food during chewing. The changes in the food structures during chewing strongly influence the release of the compounds, therefore affecting the perception of food. Here, the relationship between retronasal aroma and food deliciousness based on the physicochemical properties of aroma compounds was examined. We considered the consumption of solid foods and the effect of oral parameters in elderly people. Beef pate was used as a model food sample to study the effect of the release of aroma compounds under controlled in vitro mastication and salivation conditions using a chewing simulator. We identified the effects of coexisting ingredients such as beef fat on the time course behavior of the release of aroma compounds. In particular, the release of the middle types of aromas was significantly faster with stronger chewing force, and higher with a high fat content of the sample. In addition, a larger release intensity was observed when soy proteins were partially substituted for beef proteins. Using an appropriate model saliva, a change in the salting-out effect from the saliva composition was found to be a factor, which could explain the lowering of aroma sensation in an elderly person.

## 1. Introduction

Odors are complex stimuli, which are produced by mixtures of volatile components or compounds with various chemical structures and properties [1]. Each volatile component displays a different degree of volatility. Therefore, depending on the volatility and amount of each component in a mixture, a unique and precise fragrance is produced by an odor item. Unique fragrances also result from the quasi-infinite combinations of odorant compounds, which can be perceived as particular fragrances by our highly sensitive olfactory system. Odorant volatile compounds reach the olfactory epithelium by two different pathways, including the orthonasal and retronasal pathways, which are activated by sniffing directly the food and when the food is subjected to complex oral processing, respectively [2].

In particular, orthonasal aroma is caused by the gas phase in the headspace of food that is being aspirated from the nasal cavity, resulting in a phenomenon in which fresh air dilutes the gas phase. In the case of retronasal aroma, the targeted aroma compounds are released from food during chewing. The changes in the food structures during chewing, including crushing, grinding, disintegration, melting, dilution by saliva, etc., strongly influence the release of the compounds, thus affecting the perception of the food [3,4,5].

During chewing and swallowing, the release of aroma compounds is dependent on the composition and texture of food, saliva and mastication parameters, and the nature of the aroma compounds [6]. For example, in foods that contain fat, the retention and release of aroma compounds mainly depends on their solubility in the oil phase or hydrophobicity [7,8]. Aroma compounds are typically more soluble in fat than water. Consequently, the release of aroma compounds in the mouth from low-fat foods was found to be higher than regular-fat foods [8]. Meanwhile, proteins are able to interact with aroma compounds according to their physicochemical properties; therefore, they influence the availability of aroma compounds during all stages in the oral processing of food [7]. The introduction of other protein sources into food, such as legume proteins, can potentially modify the overall availability of certain volatile compounds, thus changing the global aroma properties of the food [9,10,11].

During oral processing of food, the release of aroma compounds varies according to oral parameters [12]. These parameters are highly dependent on individual oral physiologies [3,4,5], which lead to high degrees of variability. To overcome the constraints on aroma release studies in human subjects, chewing devices have been designed to simulate in vitro some functionalities of the oral processing of food. In particular, current devices with high functionalities are now able to mimic the compression and shearing of teeth and the functions of tongue and saliva [13,14]. They are capable of realistically reproducing the mechanical breakdown of food and producing the relevant aroma release kinetics under controlled conditions, hence enabling us to reproduce in vitro a large variety of human mastication profiles. The main advantage of these models is the ability to isolate a single parameter and study its influence on food breakdown and volatile compound release. The characteristics of the different existing devices developed in the last thirty years and associated research results are largely developed in [13,14]. For example, by controlling and decoupling the bite force, shearing angle and salivary flow rate, using a chewing simulator connected in-line with a mass spectrometer, it was possible to evaluate the impact of individual oral and food composition parameters, and the interactions between each of them, on the release of each volatile compound [15].

In this study, we firstly examined the effect of meat pate composition and some selected oral parameters on the release of aroma compounds under controlled oral conditions using a chewing simulator, with the objective of healthier food development. We also investigated the effect of fat content on aroma release, as low-fat foods are promoted to avoid obesity. Additionally, since, in recent years, alternatives to meat using plant proteins have been popularized in Europe and North America for public health reasons, we investigated the release of pate aroma compounds when replacing a part of the beef used for pate with soy protein.

Our second objective was related to a better understanding of the involvement of some oral parameters affecting flavor release and perception in elderly people, using an in vitro approach. Indeed, in the current aging society, the number of elderly people who cannot enjoy eating due to the deterioration of their eating function has been increasing. A decrease in the ability to masticate, weakness of swallowing muscles, and changes in saliva composition and secretion volume contribute to a decrease in taste, smell, and other perceptions. In this study, the effect of oral parameters on the release of aroma compounds under controlled chewing and saliva conditions, which considered the eating of an elderly person, was also examined with the chewing simulator.

## 2. Results

### 2.1. Effects of the Strength of Mastication and the Amount of Fat on Aroma Release

The release of aroma compounds from pate samples was investigated using proton transfer reaction mass spectrometry (PTR-MS). The samples with different fat contents (0, 10, and 20%) were examined at two (weak and normal) mastication strengths. The values for the maximum peak intensity of a volatile compound in the gas phase measured by PTR-MS (Imax) and the corresponding time when Imax was reached (Tmax) are shown in Figure 1. Significant differences in Tmax values were observed for butanoic acid, furfural, and dimethyl trisulfide (DMTS). The release of these three compounds was significantly faster at the highest lipid content of 20%. At this fat value, the Tmax values were significantly lower in the normal than the weak mastication strengths. The Imax results show that the released amounts of butanoic acid and furfural were significantly higher at the normal mastication strength. The effect of fat content was more significant. In general, the released amounts were lower when the lipid contents were higher. However, no significant differences in the amounts were observed for methanethiol (MeSH). Meanwhile, the release of the aroma compounds was faster with normal mastication strength and slower with 10–20% lipid contents, as shown in Figure 2.

Based on the results in Figure 1 and Figure 2 and data reported in Table 1, the release of the studied aroma compounds from the pate sample can be classified into three types: the “top type” (Log Kaw (log of air–water partition coefficient < 0) such as MeSH, which volatilized rapidly in the initial stage and gradually decayed; the “last type” (Log Kow (log of n-octanol–water partition coefficient) > 3) such as nonanal, which volatilized gradually during chewing and increased rapidly before and after swallowing; and the “middle type” (Log Kaw > 0 and 0 < Log Kow < 2) such as the organic acid DMTS and furfural, which exhibited intermediate behaviors of the top and last types. The release pattern of each studied aroma compound is presented in Figure 2, taking into account this classification.

### 2.2. Influence of Soy Protein on the Release of the Aroma Compounds

The effect of the addition of soy protein to the pate sample on the release of the aroma compounds (Tmax and Imax values) are shown in Figure 3 and Figure 4. In Figure 3, the addition of soy protein in pate with and without fat is compared. As previously observed, the mastication strength produced minimal effects on the release of the aroma compounds. However, in the pate sample with fat, the addition of soy proteins produced significantly lower Tmax values for butanoic acid, furfural and DMTS and significantly higher values for Imax for DMTS and nonanal in normal compared to weak mastication. Therefore, mastication, or chewing, has a significant effect on the release of certain aroma compounds. However, the above phenomenon was not observed in the case of pate without fat.

### 2.3. Changes in the Release of the Aroma Compounds Due to the Differences in Saliva Composition and Flow at Different Chewing Times

Three salivation conditions were used, including one standard that represented an adult and two variations in composition and flow (Table 2), which were more representative of the salivation conditions in the elderly. In the two salivary conditions in the elderly, the release of the aroma compounds was delayed due to the increase in the Tmax values as shown in Figure 5. Significantly lower Imax was also observed in elderly salivation for furfural, DMTS and nonanal. A decrease in Imax values was also observed, thus confirming the tendency of the aroma release intensity to be suppressed.

## 3. Discussion

This work aimed to study the effect of meat pate composition and some selected oral parameters on the release of aroma compounds under controlled oral conditions using a chewing simulator. This is the first in vitro study on the release of aroma compounds from meat products.

Chewing devices were designed to simulate some functions of oral food processing in vitro, mainly in order to overcome in vivo constraints such as the variability in the individuals. In particular, this chewing simulator can mimic tooth compression and shearing, as well as tongue and saliva functions. It can reproduce the mechanical breakdown of food and generate relevant aroma release kinetics under controlled conditions, allowing in vitro reproduction of a wide variety of human mastication profiles with high reproducibility. In addition to the above advantages, this device enables a measurement time of about 5 min, including cleaning of the device, compared to in vivo studies, which leads to a reduction in experiment time. In addition, there is no need to recruit volunteers. On the other hand, it is hoped that missing functionalities are progressing and may be implemented in the system in the near future, such as the reproduction of the oral mucosa and swallowing, which affects the aroma release behavior in the oral cavity. The mechanical mastication increased the release of volatile aroma compounds during the oral processing of food, as expected [16,17]. The transformation of the beef pate sample, which was caused by the combined action of mastication and salivation in the chewing simulator can increase the evaporation of the volatile compounds to the gas phase. This was achieved by increasing the surface area of the sample through an increase in bolus hydration, particle formation, and particle softening. However, the effect of mastication was different depending upon the fat content and nature of the volatile compounds. For butanoic acid, furfural, and DMTS, the Tmax values were smaller when the fat content in the beef pate sample was highest. This was likely due to the softer texture of a high-fat content product, leading to an earlier release of these volatile compounds [18]. To reach the same bolus stage, a larger number of chews was required in weak than in normal mastication; thus, higher Tmax values were observed in the weak mastication condition. With regards to the maximum release intensity, the slower release of butanoic acid, furfural, DMTS, and nonanal in the presence of fat was most likely due to the higher retention of these compounds with hydrophobic characteristics (Log P (log Kow) > 1) [7,8]. This phenomenon has been reported in studies that used model emulsions [19] and dairy products [20].

In recent years, alternative meats using plant proteins have been popularized in Europe and North America. As a reference experiment, the release of pate flavors was investigated by replacing part of the beef used for pate with soy protein. The effect of this partial substitution differed according to the nature of the volatile compound. Among the five volatile compounds studied, MeSH, butanoic acid, and DMTS showed a lower Tmax when soy proteins were added. This can be explained by the softer texture observed in the partially substituted beef pate, which allowed the earlier release of these volatile compounds. As reported previously, weaker mastication results in a higher Tmax value due to a delay in the release process [16]. This was observed in the partially substituted beef pate, which was likely due to particular texture properties that have not been studied in detail. Concerning the Imax, the larger values observed for DMTS and nonanal in partially substituted beef pate suggest differences in the retention properties between meat and soy proteins. High-temperature treatment such as cooking generally leads to an increase in the hydrophobicity of proteins, which are more capable of retaining aroma compounds [21,22]. Soy protein appeared to be less conducive to aroma retention, which was possibly due to its lower hydrophobicity. However, a decrease in the surface hydrophobicity of meat proteins may be due to the addition of soy protein [22].

With the current aging society, the number of elderly people who cannot enjoy eating due to the deterioration of their eating function has been increasing. A decrease in the ability to masticate, weakness of swallowing muscles, and changes in saliva composition and secretion volume all contribute to a decrease in taste, smell, and other perceptions [23,24]; however, these seem to involve much more complex mechanisms [25]. In this study, the effect of oral parameters on the release of aroma compounds under controlled chewing and saliva conditions, which considered the eating of an elderly person (See Table 2 conditions 2 and 3), was also examined with the chewing simulator. If an increase in the number of bites leads to an increase in the release of aroma compounds (except for MeSH due to its high volatility), the effects of saliva composition and flow provided in this study were notable. Overall, changes in the standard formulation and flowrate of artificial saliva typically resulted in a decrease in the release of aroma compounds. The decrease was most likely due to the decrease in salivary flow rate, rather than any change in the concentration of mucin or mineral salts. However, the decrease in salivary flow also contributed to a decrease in the concentration of mineral salts during the mixing of bolus and saliva in the chewing process. This decrease in concentration of mineral salts (including inorganic salts contained in the artificial saliva) could be responsible for the decrease in the salting-out effect. However, the effect of saliva on the release of aroma compounds is complex and sometimes contradictory [26]. For example, it has been reported that the dilution effect of saliva decreased the viscosity of a product and contributed to a more intense perception [27,28]. Therefore, a decrease in salivary flow rate could be a cause, which makes it difficult to sense the aroma of food in elderly people.

In conclusion, through the presented in vitro study on beef pate, we were able to control certain oral parameters that were varied independently by the chewing simulator. In particular, we were able to explain the effect of fat and soy protein contents on the temporal release of aroma compounds. This is important in the better reformulation of healthy foods with less fat and for the partial substitution of animal with plant proteins. In addition, the formulation of artificial saliva with varying levels of proteins or minerals made it possible to mimic the composition of saliva in an elderly person. We were also able to understand the impacts of formulation variations on aroma release and factors that contributed to a loss of perception. The effects of substitution of meat protein by different protein sources such as legumes, plants, and microorganisms could be determined by future aroma release studies in order to optimize the perception of new and healthier foods. Meanwhile, to better understand the release of aroma compounds in elderly people, other relevant parameters could be implemented in the programming of a chewing simulator. These will allow us to follow the release kinetics of aroma compounds with more precision. Finally, this work is expected to contribute to the better formulation of foods with a perception that is directed to the elderly population.

## 4. Materials and Methods

### 4.1. Materials

Beef pate was prepared as follows. Minced lean beef meat (50 g) was placed in a mold (80 × 60 × 8 mm) and shaped. It was then grilled on a 180 °C hot plate for 3 and 2 min on the front and back sides, respectively. Fried beef pate that was mixed with the aroma compounds was treated similarly. High-fat samples with 10% and 20% added bovine fat were also treated in the same way.

Partially substituted beef pate was prepared as follows. Minced lean beef meat (35 g) and rehydrated soybean protein “New Fuji Nick” AR from Fuji Oil (Osaka, Japan) (15 g) were mixed, placed in a mold, and shaped. The grilling method used was the same as for the beef pate.

Each aroma compound from Merck KGaA(Darmstadt, Germany) was added to the samples at a final concentration of 0.1% (1000 ppm).

All inorganic compounds of high purity were obtained from Merck. The standard artificial saliva was prepared by dissolving NaHCO_3_ (0.397 g), K_2_HPO_4_ (0.645 g), NaCl (0.067 g), KCl (0.774 g), CaCl_2_ (0.205 g), and mucin from porcine stomach (M1778–10 g) obtained from Merck (2.16 g) [29] in 1 L of purified water (from a milli-Q system from Merck Millipore). The artificial saliva specimens were formulated on the basis of those used in previous works [5,30]. The formulations used in this work are summarized in Table 2.

### 4.2. PTR-Time-of-Flight (ToF)-MS

A PTR-ToF-MS 8000 from Ionicon Analytik (Innsbruck, Austria) was used. It was connected to the chewing simulator device via a polyether ether ketone capillary (dimension of 1 mm with internal and external diameters of 1 m and 1/16 in, respectively), which was inserted in a heated transfer line that was maintained at 110 °C. The drift tube was kept under controlled conditions of pressure (2.3 mbar), temperature (50 °C), and voltage (480 V). The resulting field density ratio (E/N) was 110 Td. E and N were the electric field strength and gas number density, respectively (1 Td = 10^–17^ cm^2^·V). The detected aroma compounds and their ions are summarized in Table 1.

### 4.3. Chewing Simulator

The chewing simulator connected to the PTR-ToF-MS apparatus was specifically developed for studying the in vitro release of aroma compounds [15,31]. The samples used were 5 g pate pieces. The operating conditions included force and shear angle of 25 daN and 2°, respectively, unless otherwise stated. The influx of artificial saliva into the chewing simulator was carried out at a setting of 1 mL/min. The data were collected in five replicates (*n* = 5) and averaged. Nitrogen flow to the simulator was at 70 mL/min. For weak mastication, the chewing simulator operation conditions for force and shear angle were 5 daN and 2°, respectively. For normal mastication, these were 30 daN and 3°, respectively. The main ions for each volatile compound were checked using reference single-aroma compound solutions. Data were collected for ions that corresponded to protonated aroma compounds (see Table 1). The intensity of a signal was expressed in arbitrary units. The air exhaled by the empty and clean chewing simulator was recorded. This was subtracted from the measurements obtained when chewing the pate samples.

### 4.4. Data Processing

PTR-MS aroma compounds release curves were processed in the following manner. Slope of the *n*th mastication was the relative peak intensity from the *n*th mastication divided by the peak intensity from the 1st mastication. The Imax value was the maximum peak intensity. The Tmax was the number of chews (mastication) that showed a maximum peak intensity.

### 4.5. Statistical Analysis

The results were expressed as a mean of at least five measurements (*n* = 5). Statistical analysis was performed using Rstudio 1 September 2021 build 372 from Rstudio, PBC (Boston, MA, USA). Analyses of variance and the Tukey–Kramer method were performed at the level of α = 0.05 perform multiple comparisons. The tests were performed independently for each aroma compound. The variation in aroma compound release parameters was achieved using a model that included product, in vitro oral parameter, and product * in vitro oral parameter.

## Figures and Tables

**Figure 1 molecules-27-03259-f001:**
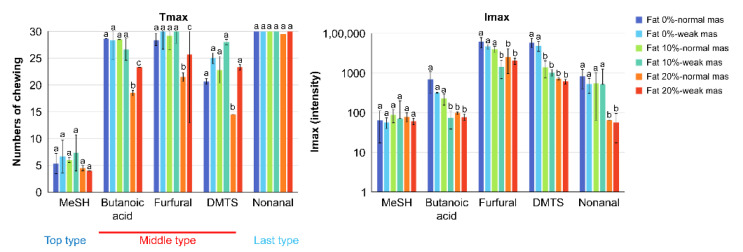
Changes in the release of aroma compounds caused by differences in the mastication (mas) strengths (normal and weak) and in the amounts of fat (Tmax and Imax). Statistical analyses were performed independently for each aroma compound. Different letters on top of bars mean significant differences (α < 0.05).

**Figure 2 molecules-27-03259-f002:**
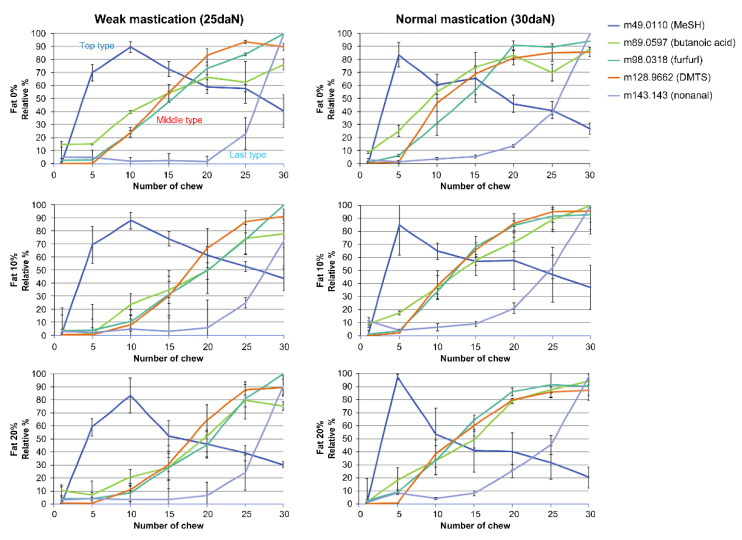
Changes in the release of aroma compounds caused by differences in mastication strengths and amounts (%) of fat (release curve). The number of chews means the number of cycles of chewing simulator jaw.

**Figure 3 molecules-27-03259-f003:**
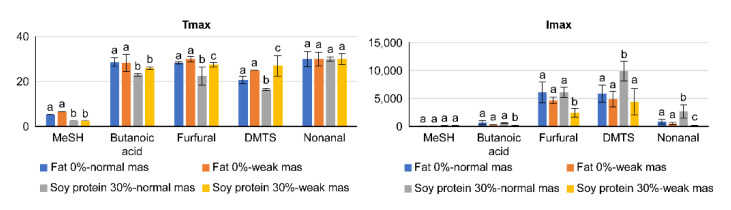
Changes in the release of aroma compounds due to differences in the strengths of mastication and in contents of soy protein (Tmax and Imax values). Statistical analyses were performed independently for each aroma compound. Different letters on top of bars mean significant differences (α = 0.05).

**Figure 4 molecules-27-03259-f004:**
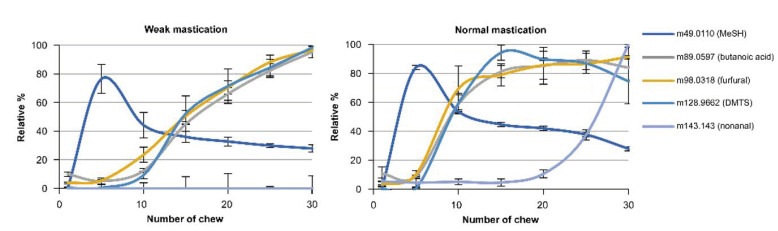
Changes in the release of aroma compounds due to differences in the strengths of mastication and in content of soy protein (release curve). The number of chews means the number of cycles of the chewing simulator jaw. Subfigure represents scanning ions measured by PTR-MS.

**Figure 5 molecules-27-03259-f005:**
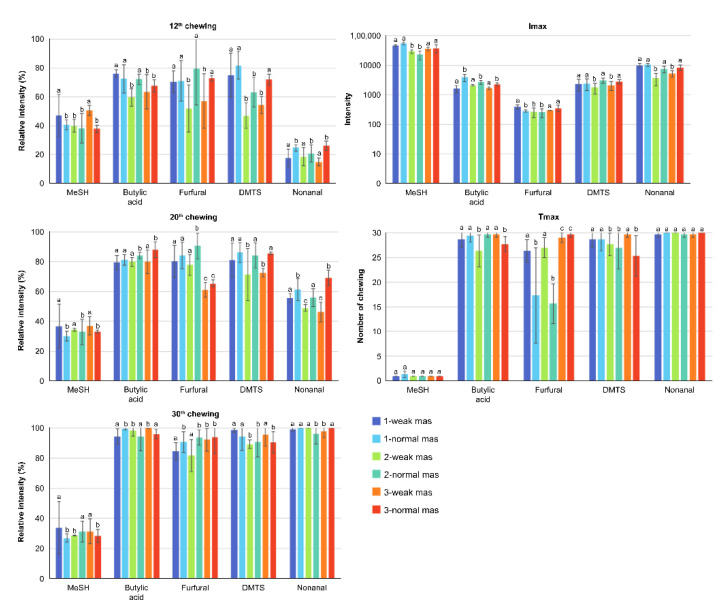
Changes in the release of aroma compounds due to differences in the strengths of mastication and in saliva composition and flow. Statistical analyses were performed independently for each aroma compound. Different letters on top of bars mean significant differences (α = 0.05).

**Table 1 molecules-27-03259-t001:** The studied aroma compounds PTR-MS *m*/*z*, Log Kow, and Log Kaw values. The value of Log Kow and Log Kaw are calculated by EPI suite (https://www.epa.gov/tsca-screening-tools/epi-suitetm-estimation-program-interface, (25 April 2022)).

Compound Name	*m*/*z*	Log Kow	Log Kaw
MeSH	49.0110	0.78	−0.89
butanoic acid	89.0597	0.79	4.66
furfural	98.0318	0.41	3.81
DMTS	128.9660	1.87	1.12
nonanal	143.1430	4.79	1.52

**Table 2 molecules-27-03259-t002:** Salivation conditions tested. Standard adult (1) and elderly artificial saliva compositions and flows (2 and 3).

	Standard Adult	Elderly	
Composition (g)	(1). Std	(2) (130% Concentration, 50% Saliva Flow from Condition (1))	(3) (50% Mucin from Condition (2))
Water	1000	1000	1000
NaHCO_3_	0.397	0.567	0.567
K_2_HPO_4_	0.645	0.921	0.921
NaCl	0.067	0.0957	0.0957
KCl	0.774	1.11	1.11
CaCl_2_	0.205	0.293	0.293
Mucin	2.16	3.09	1.54
Salivary flow rate	1 mL/min	0.5 mL/min	0.5 mL/min

## Data Availability

Not applicable.

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
