# Peer review of "Retronasal Aroma of Beef Pate Analyzed by a Chewing Simulator"

_molecules, 2022, doi:10.3390/molecules27103259_

Round 1
Reviewer 1 Report
The manuscript “Retronasal aroma of beef pate analyzed by a chewing simulator” that has been sent for publication in the journal “Molecules” which studies the effect of chewing on the release of volatile compounds. The objective of the study is very interesting and the design of the experiment seems appropriate, but the authors have not been able to capture it adequately in the article. It is difficult to understand the different experiences developed with the information provided.
The manuscript needs to be thoroughly corrected so that the reader can understand the results obtained. Here are some of the things that need to be changed:
- In section 2(Results), there is a discussion of parameters that are not described until section 4. First it is necessary to define what Tmax and Imax are. Section 4.- should be section 2.
- What does "low aw" and "low ow" means
- Figures based on the number of chews are not understood until the operation of the masticator is described in section 4.
- For example in table 1 it is not indicated what log ow and log aw values mean and therefore the results cannot be evaluated. How is it measured? Which equipment gives this value?
- In figure 3, which figure is non-fat pate and which is fat pate?
- In table 2 it is not clear what each column is.
For all these reasons the article is not publishable in its current form. It needs a thorough correction to help the reader understand the results obtained.
Reviewer 2 Report
There are orders of magnitude fewer publications on retronasal perception. The research carried out by the authors is interesting and timely. The use of chewing simulators to determine retronasal aroma is an important research direction.
The abstract is too general and needs to be complemented with concrete data.
The introduction section is lacking in the presentation of the research results of chewing and saliva simulators in the literature.
The objective is too general and needs to be clarified.
The order of the chapters is not correct and should be improved.
The description of the material and method is sufficiently detailed, however, the conditional tests for the statistical analysis (ANOVA) need to be carried out in order to be feasible. Justification for the choice of post-hoc test would also be good.
For Figures 1 and Figure 4 it is necessary to add that the statistical analyses were performed separately for each component, as it is confusing, as if they were performed together for all components at the same time.
Table 2 should be re-edited, the labelling, header is confusing.
In the results sections it is useful to show significant differences more clearly.
The conclusion section is adequate.
Reviewer 3 Report
The originality of the study and the novelty it brings in the field is of actuality. The purpose of the article and its significance are stated clearly. The paper is well structured, the abstract is concise and in the topic; the introduction is supported by well-selected bibliographic data. The Experimental and Modeling Approach correctly.
In my opinion, the conclusions are too general.
I suggest including information regarding advantages, limitation of the device, number of samples that could be analyzed/interval of time, costs, etc., this will increase the manuscript value.
Reviewer 4 Report
I recommend manuscript publication in this form.
Round 2
Reviewer 1 Report
The article is now publishable in its current state.
Reviewer 2 Report
I accept the authors' replies. The abstract and introduction sections have been expanded. In the results section, the captions of tables, figures, tables, information under figures have been added and made more precise. The discussion section has also been expanded with relevant sections. Overall, the quality of the manuscript has improved a lot, I recommend the acceptance of the article in this international journal.